# Transvaginal tru-cut biopsy versus ascitic fluid cytology in gynaecologic tumours: A comparative diagnostic study

Munachiso Iheme Ndukwe[1,2,3], Petra Bretova[1,3], Ivan Praznovec[1,3], Denisa Pohanková[2], Dominik Karasek[1,3], Martin Stepan[1,3], Dominik Habes [1,3], Jan Laco[4], Eva Hovorkova[4], Akaninyene Eseme Ubom [5,6*], Milan Vosmik [2], Igor Sirak[2]

**1** Department of Obstetrics and Gynaecology, University Hospital Hradec Králové, Hradec Králové, Czech Republic, **2** Department of Oncology and Radiotherapy, University Hospital Hradec Králové, Hradec Králové, Czech Republic, **3** Department of Obstetrics and Gynaecology, Faculty of Medicine in Hradec Králové, Charles University, Hradec Králové, Czech Republic, **4** The Fingerland Department of Pathology, Charles University, Faculty of Medicine in Hradec Králové, University Hospital Hradec Králové, Hradec Králové, Czech Republic, **5** Department of Obstetrics and Gynaecology, Faculty of Clinical Sciences, PAMO University of Medical Sciences, Port Harcourt, Nigeria, **6** Department of Obstetrics and Gynaecology, Faculty of Clinical Sciences, PAMO Teaching Hospital, Port Harcourt, Nigeria

* aubom@pums.edu.ng, bedom2001@yahoo.com

## Abstract

### Objective

To compare the diagnostic performance- including adequacy, accuracy, safety, sensitivity, specificity, and predicitve values- of transvaginal tru-cut biopsy and ascitic fluid cytology in women with gynaecologic tumours.

### Methods

A retrospective review was conducted of women with gynaecologic tumours who underwent ultrasound-guided transvaginal tru-cut biopsy and ascitic fluid cytology at the Department of Obstetrics and Gynaecology, University Hospital Hradec Kralove, between January 2018 and December 2021.

### Results

A total of 104 women with gynaecologic tumours underwent transvaginal tru-cut biopsy, of whom 47 also had ascitic fluid cytology. The diagnostic accuracy [93.3% (95% CI: 84.1%−97.4%) vs. 83.9% (95% CI: 67.4%−92.9%)], sensitivity [93.3% (95% CI: 82.1%−97.7%) vs. 79.2% (95% CI: 59.5%−90.8%)], and negative predictive value [82.4% (95% CI: 59.0%−93.8%) vs. 58.3% (95% CI: 32.0%−80.7%)] were higher for transvaginal tru-cut biopsy compared to ascitic fluid cytology. Adequacy [93.3% (95% CI: 86.8%−96.7%) vs. 93.6% (95% CI: 82.8%−97.8%)], specificity [93.3% (95% CI: 70.2%−98.8%) vs. 100.0% (95% CI: 64.6%−100.0%)], and positive predictive value [97.7% (95% CI: 88.2%−99.6%) vs. 100.0% (95% CI: 83.2%−100.0%)] were similar

**Data availability statement:** All relevant data are within the manuscript.

**Funding:** This work was financially supported by MH CZ–DRO (UHHK, 00179906), the Cooperatio Program of Charles University (ONCO, MATC, DIAG), the Grant Agency of Charles University (GAUK, Project No. 246125, Faculty of Medicine, Hradec Králové), and project BBMRI-CZ LM2023033. The funders had no role in study design, data collection and analysis, decision to publish, or preparation of the manuscript.

**Competing interests:** The authors have declared that no competing interests exist.

for the two methods. Only one tru-cut biopsy was complicated by bleeding, giving an overall complication rate of 1% (95% CI: 0.2%−5.3%). There was no complication with ascitic fluid cytology. Concordance of tru-cut histopathological diagnoses with ascitic fluid cytological diagnoses was 78% (95% CI: 63.6%−87.5%).

## Conclusion

Ultrasound-guided tru-cut biopsy provides more diagnostic information than ascitic fluid cytology and is the preferred method when feasible. However, ascitic fluid cytology remains a safe and practical option when biopsy is unavailable.

## Introduction

Ascites, defined as the pathological collection of fluid within the peritoneal cavity, is malignant in 10% of cases, with malignant ascites frequently complicating gynecologic cancers, especially ovarian cancers [1,2]. Thirty percent of patients with ovarian cancer have ascites at diagnosis and over 60% at the time of death [3]. The etiology of malignant ascites is multifactorial. In case of peritoneal carcinomatosis, fluid secretion surpasses fluid absorption by the peritoneum due to increased fluid production by tumour cells, increased vascular permeability accompanied by increased vascular hydrostatic pressure with decreased plasma oncotic pressure, release of inflammatory cytokines, tumour obstruction of lymphatic drainage and increased portal pressure from tumour metastasis [1,4].

Characteristics that differentiate malignant from benign ascites include positive cytology with a white blood count typically > 500 leucocytes/mm$^3$, high ascitic fluid protein concentration of ≥ 2.5 g/dL, and a low serum-albumin ascites gradient (SAAG) < 1.1 g/dl [2,4,5]. The high protein content of malignant ascitic fluid is due to increased vascular permeability that allows large molecules to accumulate in the intraperitoneal space, as well as secretion of proteinous fluid by tumour cells [4,5]. Cytopathological examination of ascites has an overall sensitivity of 50–96.7% for the detection of malignant ascites [6].

Tru-cut biopsy is a simple, safe, cost-effective, minimally invasive technique to obtain tissue specimens for histopathological analyses [7,8]. Compared to fine needle aspiration biopsy (FNAB), which produces small, often fragmented tissue samples that lack structural integrity, tru-cut biopsy yields larger, architecturally preserved tissue cores that allow for additional histopathological analyses, including immunohistochemistry [8–11]. Tru-cut biopsy is particularly useful for making preoperative diagnoses to plan adjuvant treatment in patients with advanced or inoperable tumours or who are unfit for surgery owing to comorbidities, eliminating the risk of morbidity and mortality from surgery [8–10]. It has a diagnostic accuracy of 76%−99% [7,8,12,13]. This study sought to compare the diagnostic performance- including accuracy, adequacy, safety, sensitivity, specificity, and predictive values- of tru-cut biopsy and ascitic fluid cytology in women with gynaecologic tumours.

## Materials and methods

This was a retrospective study of women with gynaecologic tumours who underwent ultrasound-guided transvaginal tru-cut biopsy and ascitic fluid cytology

in the Department of Obstetrics and Gynaecology, Charles University, Hradec Kralove, between January 2018 to December 2021 [8].

Relevant data were obtained and recorded from the women's case records, including route and site of tru-cut biopsy and ascitic fluid cytology, complications of tru-cut biopsy and ascitic fluid cytology, histopathological diagnoses of tru-cut biopsies, ascitic fluid cytology results, and postoperative histopathological diagnoses. The case records were assessed by DK and MN between November 11-December 6, 2024. DK and MN had access to information that could identify individual patients during data collection. The data were anonymized after data collection.

### Procedures for tru-cut biopsy and ascitic fluid cytology

The procedure for tru-cut biopsy in our institution has been previously described [8]. To perform a transvaginal tru-cut biopsy, the woman is placed in the lithotomy position and the biopsy is done under ultrasound guidance with a GE Voluson E8 machine (BT13). A Palium® biopsy gun is utilised, with a disposable 14–20 G tru-cut needle connected to the ultrasound probe via a needle guide. One or two biopsy samples are typically collected, but in a few cases, a third and even fourth sample is obtained, depending on the operator's clinical judgment.

For ascitic fluid cytology, in patients with ascites, a 20 ml sample of ascitic fluid was obtained via bedside paracentesis from the dependent part of the abdomen, typically in one of the lower flanks, using a 20 ml syringe and needle, according to our institutional protocol.

### Analysis of adequacy, accuracy, safety, sensitivity, specificity, positive predictive value, and negative predictive value of tru-cut biopsy and ascitic fluid cytology

**Adequacy.** A tru-cut biopsy or ascitic fluid sample was considered adequate if it was sufficient to allow histopathological examination, including immunohistochemistry [2,7,8].

**Accuracy.** Tru-cut biopsy and ascitic fluid cytology were considered accurate if there was agreement/concordance of tru-cut biopsy histopathological diagnosis and ascitic fluid cytology with definitive postoperative histopathological diagnosis in women who had tru-cut biopsy or ascitic fluid cytology and subsequently underwent surgery [2,7,8]. That is, the proportion of correct tru-cut biopsy and ascitic fluid cytology diagnoses, calculated as [True Positive (TP) + True Negative (TN)]/Total number of adequate/diagnostic tru-cut biopsy or ascitic fluid cytology samples. True positives were women with malignancy (positive postoperative histopathological diagnosis) who had positive tru-cut biopsy or ascitic fluid cytology, while TNs were women without malignancy (negative postoperative histopathological diagnosis) who had negative tru-cut biopsy or ascitic fluid cytology. Women who had inadequate/non-diagnostic tissue samples on tru-cut biopsy and ascitic fluid cytology were excluded from evaluation for accuracy.

**Safety.** Safety was assessed based on the presence or absence of complications following tru-cut biopsy and ascitic fluid cytology [2,7,8].

**Sensitivity.** This was defined as the probability that a woman with malignancy will have a positive tru-cut biopsy or ascitic fluid cytology. It was calculated as: [TP/(TP + False Negative (FN))] x 100% [14]. False negatives were women with malignancy (positive postoperative histopathological diagnosis) who had negative tru-cut biopsy or ascitic fluid cytology.

**Specificity.** This was defined as the probability that a woman without malignancy will have a negative ascitic fluid cytology or tru-cut biopsy. It was calculated as: [TN/(TN + False Positive (FP))] x 100% [14]. False positives were women without malignancy (negative postoperative histopathological diagnosis) who had positive tru-cut biopsy or ascitic fluid cytology.

**Positive Predictive Value (PPV).** This was defined as the probability that a woman with a positive tru-cut biopsy or ascitic fluid cytology truly had a malignancy (positive postoperative histopathological diagnosis). It was calculated as: [TP/(TP + FP)] x 100% [14].



**Negative Predictive Value (NPV).** This was defined as the probability that a woman with a negative ascitic fluid cytology or tru-cut biopsy truly did not have a malignancy (negative postoperative histopathological diagnosis). It was calculated as: [TN/(TN+FN)] x 100% [14].

## Statistical analysis

Data obtained were analysed using the IBM SPSS Statistics version 25 (IBM Corp., Armonk, N.Y., USA). Adequacy, accuracy, safety, sensitivity, specificity, PPV, and NPV, were determined and compared for tru-cut biopsy and ascitic fluid cytology. Given the small sample sizes, 95% confidence intervals (CI) for adequacy, accuracy, sensitivity, specificity, PPV and NPV, were calculated using the Wilson score interval method.

## Ethics

This study was approved by the Ethical Committee of University Teaching Hospital, Hradec Kralove, with a reference number of 202411 P05, on 10 October 2024. Informed consent was waived due to the retrospective nature of the study and anonymization of patient data.

## Results

Within the study period, 104 women with gynaecologic tumours underwent tru-cut biopsy, out of which 47 also had ascitic fluid cytology.

### Baseline characteristics of the study participants

The ages of the women ranged from 26 to 84 years, with a mean age of 61.6 ± 12.1 years. They were predominantly post-menopausal (86, 82.7%), and overweight, with a mean body mass index (BMI) of 27.0 ± 6.4 kg/m² (Table 1).

### Sensitivity, specificity, PPV and NPV of tru-cut biopsy versus ascitic fluid cytology

Sixty-four (61.5%) women who underwent tru-cut biopsy, including 34/47 (72.3%) women who had both transvaginal tru-cut biopsy and ascitic fluid cytology, subsequently had surgery followed by postoperative definitive histopathological diagnoses. Of this number, 15 (23.4%) were benign tumours, while 49 (76.6%) were malignant. Advanced ovarian cancers were the predominant malignant tumours (40/49, 81.6%), with serous cystadenocarcinomas constituting more than four-fifths of ovarian cancers (33/40, 82.5%) (Table 2).

Tru-cut biopsies were inadequate/non-diagnostic in four of the 64 women who had transvaginal tru-cut biopsy and subsequently surgery. Of the 60 women with adequate transvaginal tru-cut biopsies, 42 were TP for malignancy, 14 were TN,

**Table 1. Baseline characteristics of the study participants.**

| Characteristic | Frequency, $n$=104 | Percentage (%) |
|---|---|---|
| **Age (years)** | | |
| <40 | 4 | 3.8 |
| 40-49 | 17 | 16.4 |
| 50-59 | 23 | 22.1 |
| ≥60 | 60 | 57.7 |
| **Menopausal status** | | |
| Premenopausal | 18 | 17.3 |
| Postmenopausal | 86 | 82.7 |
| **Mean BMI (kg/m²)** | 27.0 ± 6.4 | |



**Table 2.** Postoperative histopathological diagnoses of women who underwent transvaginal tru-cut biopsy and ascitic fluid cytology.

| Characteristic | Frequency | Percentage (%) |
|---|---|---|
| **Postoperative histopathological diagnosis (*n*=64)** | | |
| Malignant | 49 | 76.6 |
| Benign | 15 | 23.4 |
| **Type of malignancy (*n*=49)** | | |
| Advanced ovarian cancer | 40 | 81.6 |
| Metastases from non-genital cancer | 6 | 12.3 |
| Advanced endometrial cancer | 3 | 6.1 |
| **Histological subtype of ovarian cancer (*n*=40)** | | |
| Serous cystadenocarcinoma | 33 | 82.5 |
| Endometrioid carcinoma | 2 | 5.0 |
| Mucinous cystadenocarcinoma | 1 | 2.5 |
| Clear cell carcinoma | 1 | 2.5 |
| Malignant rhabdoid tumour | 1 | 2.5 |
| Adenoid cystic carcinoma | 1 | 2.5 |
| Leiomyosarcoma | 1 | 2.5 |

three were FN, and one was a FP; tru-cut biopsy had a sensitivity of 93.3% (95% CI: 82.1%−97.7%), specificity of 93.3% (95% CI: 70.2%−98.8%), PPV of 97.7% (95% CI: 88.2%−99.6%), and a NPV of 82.4% (95% CI: 59.0–93.8%) (Table 3).

Thirty-four (72.3%) of the 47 women who had ascitic fluid cytology went on to have surgery with definitive postoperative histopathological diagnoses. Of this number, three had non-diagnostic/inadequate ascitic fluid samples. Nineteen of the 31 operated women with adequate ascitic fluid samples were TP for malignancy, seven were TN, five were FN, and none was FP; the sensitivity, specificity, PPV, and NPV of ascitic fluid cytology were, respectively, 79.2% (95% CI: 59.5%−90.8%), 100% (95% CI: 64.6%−100.0%), 100% (95% CI: 83.2%−100.0%), and 58.3% (95% CI: 32.0%−80.7%) (Table 3).

## Comparison of adequacy, accuracy, and safety of tru-cut biopsy and ascitic fluid cytology

The overall adequacy of tru-cut biopsy was 93.3% (95% CI: 86.8%−96.7%). A non-diagnostic/inadequate sample was seen in only 6.7% (7/104) of cases. The overall adequacy of ascitic fluid cytology was 93.6% (95% CI: 82.8%−97.8%), similar to the adequacy of tru-cut biopsy (Table 4). A non-diagnostic/inadequate sample for ascitic fluid cytology was obtained in only 3/47 (6.4%) cases.

Overall accuracy of tru-cut biopsy was 93.3% (95% CI: 84.1%−97.4%) [(TP+TN)/Total adequate/diagnostic tru-cut biopsy samples; (42+14)/60], with the definitive postoperative histopathological diagnoses in 56 of the 60 operated women who had adequate/diagnostic tru-cut biopsies, being in agreement with the tru-cut biopsy histopathological diagnoses. Of the 31 women with adequate ascitic fluid cytology who had surgery, the accuracy of ascitic fluid cytology was 83.9% (95% CI: 67.4%−92.9%) [(TP+TN)/Total adequate/diagnostic ascitic fluid cytology samples; (19+7)/31].

Only one tru-cut biopsy was complicated by bleeding, giving an overall complication rate of tru-cut biopsy of 1% (95% CI: 0.18–5.31). There was no complication with ascitic fluid cytology (Table 4).

**Table 3.** Sensitivity, specificity, PPV, and NPV of tru-cut biopsy versus ascitic fluid cytology.

| Diagnostic modality | Sensitivity (%) (95% CI) | Specificity (%) (95% CI) | PPV (%) (95% CI) | NPV (%) (95% CI) |
|---|---|---|---|---|
| Tru-cut biopsy | 93.3 (82.1-97.7) | 93.3 (70.2-98.8) | 97.7 (88.2-99.6) | 82.4 (59.0-93.8) |
| Ascitic fluid cytology | 79.2 (59.5-90.8) | 100.0 (64.6-100.0) | 100.0 (83.2-100.0) | 58.3 (32.0-80.7) |

**Table 4. Accuracy, adequacy, and procedure-related complications of tru-cut biopsy versus ascitic fluid cytology.**

| Characteristic | Tru-cut biopsy (95% CI) | Ascitic fluid cytology (95% CI) |
| --- | --- | --- |
| Overall accuracy (%) | 93.3 (84.1-97.4) | 83.9 (67.4-92.9) |
| Adequacy (%) | 93.3 (86.8-96.7) | 93.6 (82.8-97.8) |
| Complication rate (%) | 1.0 (0.2-5.3) | 0 |

### Concordance of tru-cut biopsy with ascitic fluid cytology

Forty-seven women underwent both tru-cut biopsy and ascitic fluid cytology. Of these, three each of tru-cut biopsy and ascitic fluid cytology had inadequate/non-diagnostic tru-cut biopsy and ascitic fluid cytology samples. In the remaining 41 cases, tru-cut biopsy histopathological diagnoses were concordant with ascitic fluid cytology in 32 (78.0%, 95% CI: 63.6%−87.5%) cases, and discordant in 9 (22.0%, 95% CI: 12.0%−36.7%) cases.

### Discussion

This study found similar adequacy rates of 93.3% and 93.6% for tru-cut biopsy and ascitic fluid cytology, respectively. The overall diagnostic accuracy of tru-cut biopsy was 93.3%. On the other hand, ascitic fluid cytology had an overall accuracy of 83.9%. Only one tru-cut biopsy was complicated by bleeding, whereas there was no complication associated with ascitic fluid cytology. Tru-cut biopsy had higher sensitivity (93.3% vs. 79.2%) and negative predictive value (82.4% vs. 58.2%) than ascitic fluid cytology, while specificity (93.3% vs. 100.0%) and positive predictive value (97.7% vs. 100.0%) for both were comparable

Cytopathology is the gold standard test for confirming malignancy in patients with ascites and a suspicion of cancer [15]. Assay of tumour markers in ascitic fluid has been proposed for the diagnosis of malignancy-related ascites, but the sensitivity and specificity of tumour markers in ascitic fluid do not demonstrate any superiority over cytological diagnosis [16,17]. Ng et al documented that a greater volume of ascitic fluid was associated with higher discordance between ascitic fluid cytology and postoperative histopathological diagnoses [18]. They reported that an ascitic fluid volume of 60–100 mL, and three sequential collections on different days of a patient's admission provided the best diagnostic accuracy of up to 99.5% [18]. Zhang et al recommended 200 mL as the optimum minimum volume for confirming malignant ascites [15]. In our study, 20 mL of ascitic fluid was collected only once for cytological diagnosis. This may possibly explain the lower diagnostic accuracy recorded in our study. However, irrespective of the volume, if malignant cells are found in an ascitic fluid sample, then the sample is regarded as adequate [15]. We recorded an ascitic fluid adequacy of over 90% despite the lower accuracy of <85%.

While some authors have reported the sensitivity of ascitic fluid cytology as being between 50–98.5%, others have reported a lower sensitivity of 50–70% [6,19]. The lower sensitivity may not be unconnected with the fact that a tumour might infiltrate the peritoneum but not shed malignant cells, resulting in a negative cytology result. The paraffin-embedded cell block method collects more cellular components, improving the sensitivity of ascitic fluid cytology and better demonstrating architectural patterns, which significantly helps in making the correct diagnosis of the primary lesion [20,21].

Tru-cut biopsy yields large, architecturally preserved tissues, with a higher diagnostic accuracy and sensitivity than ascitic fluid cytology [7–13]. Corroborating this, our study recorded a tru-cut biopsy diagnostic accuracy and sensitivity of both > 90% compared to < 85% and about 80%, respectively, for ascitic fluid cytology. Ultrasound guidance during tru-cut biopsy allows for a more precise tissue acquisition and whole-procedure control of the needle tip, and used in combination with colour Doppler imaging, avoids sampling errors by facilitating the selection of the most appropriate tissue area for biopsy while avoiding vascular injury, improving diagnostic accuracy, adequacy, and safety of tru-cut biopsy [8,9,12,13]. All the tru-cut biopsies in this study were done under Doppler ultrasound guidance. More so, the transvaginal route, which

was the case in all the tru-cut biopsies in this study, allows for the biopsy of pelvic masses that may be difficult to biopsy transabdominally, owing to proximity between the transvaginal probe and the pelvic mass, with the added advantage of reducing the risks of vascular and intestinal injuries [8,22]. These possibly explain the high diagnostic accuracy, adequacy, and safety of tru-cut biopsy in our study. Adequacy and accuracy of tru-cut biopsy are also affected by the site, origin, and heterogeneity of the tumour, as well as the skill and experience of the operator [8,9,23]. In a previous study, we documented that a skilled operator and advanced ovarian cancers had the highest tru-cut diagnostic accuracy [8]. We also documented that obesity, small tumour diameter, and a single biopsy sample were associated with lower tru-cut biopsy adequacy [8].

Ovarian cancer is the most common cause of malignant ascites in females, and ascites in a female with a pelvic mass is highly predictive of ovarian cancer [24,25]. A larger tumour load in advanced and large tumours correlates positively with the presence of ascites and diagnostic adequacy and accuracy of ascitic fluid cytology. Slightly more than 80% of the malignant tumours in this study were advanced ovarian cancers. This possibly explains the high ascitic fluid diagnostic accuracy and adequacy (comparable to tru-cut biopsy) recorded in this study.

This study provides valuable real-world data comparing transvaginal tru-cut biopsy and ascitic fluid cytology against postoperative histopathology, strengthening its diagnostic validity. It also adds underreported evidence from a Central European population. Limitations include its retrospective, single-centre design, and modest sample size (particularly for patients undergoing both procedures and surgery), and the predominance of patients with advanced ovarian cancer, which introduces the possibility of selection bias. Nonetheless, the study highlights the continued relevance of ascitic fluid cytology when biopsy is unfeasible and reinforces the high diagnostic performance of ultrasound-guided tru-cut biopsy in advanced ovarian cancer.

## Conclusion

Both tru-cut biopsy and ascitic fluid cytology are safe, cost-effective, and minimally invasive diagnostic tools that can be performed in an outpatient setting without anaesthesia. Tru-cut biopsy demonstrated higher overall diagnostic accuracy, sensitivity, and negative predictive value, making it the preferred modality when a definitive tissue diagnosis is required. However, ascitic fluid cytology retains a critical role in the diagnostic pathway. Its high specificity and positive predictive value, combined with ease of performance, absence of complications, and good adequacy rates, make it especially useful when biopsy is not feasible or in settings with limited resources. Importantly, a positive cytology result reliably confirms malignancy and can expedite treatment decisions, particularly in patients with advanced ovarian cancer where surgery is not immediately indicated. Thus, while tru-cut biopsy may be superior in diagnostic performance, ascitic fluid cytology remains a relevant and valuable first-line diagnostic option in gynaecologic oncology, especially in settings where biopsy is unavailable or unfeasible.

## Author contributions

**Conceptualization:** Munachiso Iheme Ndukwe, Denisa Pohanková, Igor Sirak.

**Formal analysis:** Munachiso Iheme Ndukwe, Dominik Karasek, Akaninyene Eseme Ubom.

**Funding acquisition:** Milan Vosmik, Igor Sirak.

**Methodology:** Munachiso Iheme Ndukwe, Ivan Praznovec, Dominik Habes, Eva Hovorkova.

**Software:** Dominik Karasek, Akaninyene Eseme Ubom.

**Supervision:** Milan Vosmik, Igor Sirak.

**Validation:** Petra Bretova, Jan Laco.

**Writing – original draft:** Munachiso Iheme Ndukwe, Petra Bretova, Dominik Karasek.

**Writing – review & editing:** Ivan Praznovec, Martin Stepan, Akaninyene Eseme Ubom.



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
