## [Decision Letter · Decision Letter 0]

19 Feb 2026

PONE-D-25-37903Transvaginal tru-cut biopsy versus ascitic fluid cytology in gynecologic tumours: a comparative diagnostic studyPLOS One

Dear Dr. Ubom,

Thank you for submitting your manuscript to PLOS ONE. After careful consideration, we feel that it has merit but does not fully meet PLOS ONE’s publication criteria as it currently stands. Therefore, we invite you to submit a revised version of the manuscript that addresses the points raised during the review process.

We look forward to receiving your revised manuscript.

Kind regards,

Manasa Varra

Academic Editor

PLOS One

Journal Requirements:

- DOI: 10.3390/diagnostics15091133

In your revision ensure you cite all your sources (including your own works), and quote or rephrase any duplicated text outside the methods section. Further consideration is dependent on these concerns being addressed.

This work was supported by MH CZ–DRO (UHHK, 00179906), the Cooperatio Program of Charles University (ONCO, MATC, DIAG), the Grant Agency of Charles University (GAUK, Project No. 246125, Faculty of Medicine, Hradec Králové), and project BBMRI-CZ LM2023033.

Additional Editor Comments:

Dear Akaninyene Eseme Bernard Ubom,

This is to inform you that the manuscript needs major revisions as indicated below if the prospective data is available with you.

1. There was no indication of the patient group in this study that proceeded to neoadjuvant chemotherapy and surgery based solely on ascitic fluid cytology. Therefore, in cases where ascitic cytology was reported as benign but tru-cut biopsy revealed malignancy and the patient was consequently managed as having a malignant disease, the clinical approach would have been markedly different had tru-cut biopsy not been performed. If management decisions were based only on ascitic cytology, a benign cytology result could have led to an entirely different treatment pathway. For this reason, the conclusions regarding the diagnostic value of ascitic cytology appear somewhat overstated. Provide the relevant data.

2. A more appropriate analytical framework would involve three distinct groups: patients who underwent isolated tru-cut biopsy, those who had isolated ascitic cytology, and those who underwent both tru-cut biopsy and ascitic cytology. Additionally, information on the final pathological diagnoses of these patients would have been highly informative. While ascitic cytology positivity is known to be relatively high in serious ovarian carcinoma, this is not necessarily the case for other tumor types. Therefore, the histological tumor subtypes of the included patients should have been specified and discussed in the manuscript.

Reviewers' comments:

Reviewer's Responses to Questions

**Comments to the Author**

1. Is the manuscript technically sound, and do the data support the conclusions?

Reviewer #1: Yes

Reviewer #2: Partly

Reviewer #3: Yes

2. Has the statistical analysis been performed appropriately and rigorously? 

Reviewer #1: Yes

Reviewer #2: I Don't Know

Reviewer #3: No

3. Have the authors made all data underlying the findings in their manuscript fully available?

Reviewer #1: Yes

Reviewer #2: Yes

Reviewer #3: Yes

4. Is the manuscript presented in an intelligible fashion and written in standard English?

Reviewer #1: Yes

Reviewer #2: Yes

Reviewer #3: Yes

5. Review Comments to the Author

Reviewer #1: Comments to Authors:

Abstract:

• The abstract is concise and readable. The objectives, methods, and findings are well presented.

• The conclusion is appropriately stated based on the findings.

Title:

• The title is precise, clear, and informative. The title reflects the objectives and design of the study appropriately.

Introduction:

• The introduction provides a good background in terms of ascites and cytology in gynecologic malignancies.

• The rationale of comparing tru-cut biopsy and ascitic fluid cytology is well described.

• Nevertheless, a little more elaboration on the clinical significance of this comparison to patient care and decision-making, particularly in low-resource or outpatient practice, would be a welcome addition to the introduction.

• The introduction might also be stronger if it discussed the existing literature on this topic or stated clearly how the present study uniquely contributes to the knowledge base.

Methods:

• The retrospective design was clearly described, with proper comments regarding inclusion criteria, procedure, and data analysis.

• Statistical methodology was well described (sensitivity, specificity, PPV, NPV).

• Approval of ethics and waiver of informed consent was properly stated.

• The potential methodological limitation of collecting a small ascitic sample volume (20 mL) was noted. Most studies and international reporting systems have suggested collecting larger volumes, generally ≥50–75 mL for serous effusions, while some ascites-specific studies noted an ideal sample volume of 200 mL when classifying malignant ascites. Noted guidance suggests that taking multiple samples on days subsequently or using cell-block preparation can enhance diagnostic sensitivity with small volumes1,2,3. Addressing a rational for sampling only 20 mL (e.g. according to institutional protocol, keeping within patient tolerance for comfort, or resource limitations) would have strengthened the methods section and clarify how this may have influenced the results.

Results:

• The results are presented neatly, with tables to compare diagnostic values of both techniques.

• There is good reporting of the complication rate, which adds to the clinical applicability of the findings.

• However, subgroup analysis (e.g., by tumour type or stage) could be informative, as high-grade ovarian cancers predominated the cohort.

Discussion:

• The discussion is excellent in contrasting the findings with other literature and highlighting the benefits of tru-cut biopsy versus ascitic cytology.

• The authors have referred to the limitation of single-center study, small sample size, and retrospective design, which is welcome.

• Additional discussion regarding the clinician's practical rule of thumb of how to utilize tru-cut and how frequently just cytology would suffice would be helpful.

Conclusion:

• The conclusion answers the research question appropriately.

• The authors summarize the key findings clearly, but they may consider adding a forward-looking statement on how these findings could guide diagnostic protocols in gynecologic oncology.

Overall Evaluation: This is a well-conducted and clinically relevant study, which compares two important diagnostic examinations in gynecologic oncology. It is well written, well-reasoned, and adds valuable evidence to the literature. With minor revision to further bolster the clinical utility and defend some of the methodological choices, the manuscript is publishable.

References:

1) https://doi.org/10.1002/cncy.22577

2) Zhang F, Feng Z, Zhang Y, Liu Z, Sun X, Jin S. Determination of the optimal volume of ascitic fluid for the precise diagnosis of malignant ascites. Saudi J Gastroenterol. 2019 Sep-Oct;25(5):327-332. doi: 10.4103/sjg.SJG_547_18. PMID: 30900607; PMCID: PMC6784429.

3) https://doi.org/10.1159/000529855

Reviewer #2: For the tru-cut biopsy procedure, was attention paid to patients’ pre-procedural platelet counts and the use of anticoagulants, particularly heparin? The Statistical Analysis section should be expanded in greater detail. Was a power analysis performed?

In addition, it is not clear whether patient data were adequately anonymized. Information that could potentially reveal patient identity (such as names or identifiable personal details) should not be retained in the archive, as such data are unnecessary for the purposes of this study.

Apart from this, there was no patient group in this study that proceeded to neoadjuvant chemotherapy and surgery based solely on ascitic fluid cytology. Therefore, in cases where ascitic cytology was reported as benign but tru-cut biopsy revealed malignancy and the patient was consequently managed as having a malignant disease, the clinical approach would have been markedly different had tru-cut biopsy not been performed. If management decisions were based only on ascitic cytology, a benign cytology result could have led to an entirely different treatment pathway. For this reason, the conclusions regarding the diagnostic value of ascitic cytology appear somewhat overstated.

In my view, a more appropriate analytical framework would involve three distinct groups: patients who underwent isolated tru-cut biopsy, those who had isolated ascitic cytology, and those who underwent both tru-cut biopsy and ascitic cytology. Additionally, information on the final pathological diagnoses of these patients would have been highly informative. While ascitic cytology positivity is known to be relatively high in serous ovarian carcinoma, this is not necessarily the case for other tumor types. Therefore, the histological tumor subtypes of the included patients should have been specified and discussed in the manuscript.

Reviewer #3: This is a clinically relevant retrospective diagnostic study comparing transvaginal ultrasound-guided tru-cut biopsy with ascitic fluid cytology in gynecologic tumours. However, I have some concerns given the small sample size as well as the definitions for the outcomes used.

Major comments

* Only 47/104 patients (45.2%) underwent both procedures, and only 34 had both procedures and surgery.

This introduces selection bias, especially since advanced ovarian cancer predominates.

* A list of full histopathological outcomes but as well of descriptive statistics of the study population is missing. This is very important to understand the study population.

* Accuracy is defined as concordance with postoperative histology, but the authors only refer to outcome benign vs malignant.

The value of trucut biopsy is not only to confirm malignancy, but to detect the origin of the primary tumor. Most patients undergoing trucutbiopsy or ascites punction will have proven metastasis on imaging, so it is not so clinically relevant to have the confirmation of malignancy. On the other hand, it is most important to have a histological diagnosis of the tumor in order to select the appropiate management. It would be very interesting to see the accuracy based on the final histopathological diagnosis (not only benign vs malignant)

* Missing confidence intervals

Especially given the small sample size, it is recommended to add confidence intervals to the performance measures

* it is not clear how complications were assessed

Minor comments

* Use either “tru-cut” or “Tru-Cut” consistently throughout

“gynecologic tumours” vs “gynaecologic tumours” → choose one spelling

* Table 2: “Safety” could be clarified as “Procedure-related complications”

6. PLOS authors have the option to publish the peer review history of their article (what does this mean?). If published, this will include your full peer review and any attached files.

Reviewer #1: **Yes:** Bayan Al Omari

Reviewer #2: **Yes:** Fatma Basak Tanoglu

Reviewer #3: No

---

## [Author Response · Author response to Decision Letter 1]

9 Apr 2026

Department of Obstetrics & Gynaecology,

Faculty of Clinical Sciences,

PAMO University of Medical Sciences,

Nigeria.

April 9, 2026.

The Editor-in-Chief,

PLOS ONE.

Dear Prof Chenette,

Responses to Editor’s and Reviewers’ Comments for Manuscript Entitled “Transvaginal tru-cut biopsy versus ascitic fluid cytology in gynecologic tumours: a comparative diagnostic study” (Manuscript ID: PONE-D-25-37903)

On behalf of my co-authors, I immensely appreciate the Academic Editor and Reviewers for the review of this manuscript. Please kindly find below a point-by-point response to the review comments. Changes are highlighted in red fonts in the revised manuscript.

Journal Requirements:

Comment 1: Please ensure that your manuscript meets PLOS ONE's style requirements, including those for file naming.

Authors’ response: Yes, we confirm that our manuscript meets PLOS ONE’s style requirements, including those for file naming. Thank you.

Comment 2: We noticed you have some minor occurrence of overlapping text with the following previous publication(s), which needs to be addressed: - DOI: 10.3390/diagnostics15091133

In your revision ensure you cite all your sources (including your own works), and quote or rephrase any duplicated text outside the methods section. Further consideration is dependent on these concerns being addressed.

Authors’ response: The referenced paper is our work, which we have cited in the index paper (Reference number 8). Duplicated text have been rephrased- Lines 67-69, Page 3; Lines 88-94, Pages 4-5; and Lines 103-119, Pages 5-6. Thank you.

Comment 3: Thank you for stating the following financial disclosure:

This work was supported by MH CZ–DRO (UHHK, 00179906), the Cooperatio Program of Charles University (ONCO, MATC, DIAG), the Grant Agency of Charles University (GAUK, Project No. 246125, Faculty of Medicine, Hradec Králové), and project BBMRI-CZ LM2023033.

Authors’ response: The financial disclosure has been amended to include that “The funders had no role in study design, data collection and analysis, decision to publish, or preparation of the manuscript.”- Lines 316-317, Page 15. Thank you.

Comment 4: We note that your Data Availability Statement is currently as follows: All relevant data are within the manuscript and its Supporting Information files.

Authors’ response: We confirm that the manuscript contains all relevant data needed to replicate the results of our study. This has been corroborated by all the three reviewers who responded “Yes” to the question “Have the authors made all data underlying the findings in their manuscript fully available?”. Thank you.

Comment 5: If the reviewer comments include a recommendation to cite specific previously published works, please review and evaluate these publications to determine whether they are relevant and should be cited. There is no requirement to cite these works unless the editor has indicated otherwise.

Authors’ response: Well noted. Thank you.

Additional Editor Comments:

Comment 1: There was no indication of the patient group in this study that proceeded to neoadjuvant chemotherapy and surgery based solely on ascitic fluid cytology. Therefore, in cases where ascitic cytology was reported as benign but tru-cut biopsy revealed malignancy and the patient was consequently managed as having a malignant disease, the clinical approach would have been markedly different had tru-cut biopsy not been performed. If management decisions were based only on ascitic cytology, a benign cytology result could have led to an entirely different treatment pathway. For this reason, the conclusions regarding the diagnostic value of ascitic cytology appear somewhat overstated. Provide the relevant data.

Authors’ response: We have acknowledged the diagnostic challenges of ascitic fluid cytology and the superiority of tru-cut biopsy over ascitic fluid cytology. In the Conclusion of the Abstract, we stated that: “Ultrasound-guided tru-cut biopsy provides more diagnostic information than ascitic fluid cytology and is the preferred method when feasible. However, ascitic fluid cytology remains a safe and practical option when biopsy is unavailable.” – Lines 43-45, Page 2. We also mentioned in the Conclusion in-text that: “Thus, while tru-cut biopsy may be superior in diagnostic performance, ascitic fluid cytology remains a relevant and valuable first-line diagnostic option in gynaecologic oncology, especially in settings where biopsy is unavailable or unfeasible.”- Lines 304-307, Page 15.

All the relevant data on sensitivity, specificity, positive predictive value, negative predictive value, accuracy, adequacy and safety of tru-cut biopsy versus ascitic fluid cytology have been provided in the Results section- Lines 180-216, Pages 9-11. Thank you.

Comment 2: A more appropriate analytical framework would involve three distinct groups: patients who underwent isolated tru-cut biopsy, those who had isolated ascitic cytology, and those who underwent both tru-cut biopsy and ascitic cytology. Additionally, information on the final pathological diagnoses of these patients would have been highly informative. While ascitic cytology positivity is known to be relatively high in serious ovarian carcinoma, this is not necessarily the case for other tumor types. Therefore, the histological tumor subtypes of the included patients should have been specified and discussed in the manuscript.

Authors’ response: The study aimed to compared tru-cut biopsy and ascitic fluid cytology. Data for patients who underwent tru-cut biopsy were analysed to determine the sensitivity, specificity, positive predictive value, negative predictive value, accuracy, adequacy and safety of tru-cut biopsy. The same was done for patients who underwent ascitic fluid cytology and then both compared, as well as concordance of tru-cut biopsy histopathological diagnoses with ascitic fluid cytology- Lines 180-223, Pages 9-11.

The final postoperative pathological diagnoses and histological subtypes have now been provided in Table 2 in the revised manuscript- Lines 177-179, Page 9. We agree that ascitic cytology positivity is relatively higher in serous ovarian carcinoma and have acknowledged this in the Discussion: “Ovarian cancer is the most common cause of malignant ascites in females, and ascites in a female with a pelvic mass is highly predictive of ovarian cancer...Slightly more than 80% of the malignant tumours in this study were advanced ovarian cancers. This possibly explains the high ascitic fluid diagnostic accuracy and adequacy (comparable to tru-cut biopsy) recorded in this study.”- Lines 276-282, Page 14. Thank you.

Reviewer 1:

Comment 1: The abstract is concise and readable. The objectives, methods, and findings are well presented.

Authors’ response: Thank you.

Comment 2: The conclusion is appropriately stated based on the findings.

Authors’ response: Thank you.

Comment 3: The title is precise, clear, and informative. The title reflects the objectives and design of the study appropriately.

Authors’ response: Thank you.

Comment 4: The introduction provides a good background in terms of ascites and cytology in gynecologic malignancies.

Authors’ response: Thank you.

Comment 5: The rationale of comparing tru-cut biopsy and ascitic fluid cytology is well described.

Authors’ response: Thank you.

Comment 6: Nevertheless, a little more elaboration on the clinical significance of this comparison to patient care and decision-making, particularly in low-resource or outpatient practice, would be a welcome addition to the introduction.

Authors’ response: We elaborated on the clinical significance of this comparison to patient care and decision-making, particularly in low-resource practice in the Conclusion as follows: “Tru-cut biopsy demonstrated higher overall diagnostic accuracy, sensitivity, and negative predictive value, making it the preferred modality when a definitive tissue diagnosis is required. However, ascitic fluid cytology retains a critical role in the diagnostic pathway. Its high specificity and positive predictive value, combined with ease of performance, absence of complications, and good adequacy rates, make it especially useful when biopsy is not feasible or in settings with limited resources. Importantly, a positive cytology result reliably confirms malignancy and can expedite treatment decisions, particularly in patients with advanced ovarian cancer where surgery is not immediately indicated.”- Lines 296-304, Pages 14-15. We thought it more appropriate to elaborate this in the Conclusion after presenting Results on the sensitivity, specificity, positive predictive value, negative predictive value, accuracy, adequacy and safety of both diagnostic modalities, and comparing the two. Thank you.

Comment 7: The introduction might also be stronger if it discussed the existing literature on this topic or stated clearly how the present study uniquely contributes to the knowledge base.

Authors’ response: We discussed existing literature on the topic in the Discussion. We added in the Discussion that: “This study provides valuable real-world data comparing transvaginal tru-cut biopsy and ascitic fluid cytology against postoperative histopathology, strengthening its diagnostic validity. It also adds underreported evidence from a Central European population.”- Lines 283-286, Page 14. Thank you.

Comment 8: The retrospective design was clearly described, with proper comments regarding inclusion criteria, procedure, and data analysis.

Authors’ response: Thank you.

Comment 9: Statistical methodology was well described (sensitivity, specificity, PPV, NPV).

Authors’ response: Thank you.

Comment 10: Approval of ethics and waiver of informed consent was properly stated.

Authors’ response: Thank you.

Comment 11: The potential methodological limitation of collecting a small ascitic sample volume (20 mL) was noted. Most studies and international reporting systems have suggested collecting larger volumes, generally ≥50–75 mL for serous effusions, while some ascites-specific studies noted an ideal sample volume of 200 mL when classifying malignant ascites. Noted guidance suggests that taking multiple samples on days subsequently or using cell-block preparation can enhance diagnostic sensitivity with small volumes1,2,3. Addressing a rational for sampling only 20 mL (e.g. according to institutional protocol, keeping within patient tolerance for comfort, or resource limitations) would have strengthened the methods section and clarify how this may have influenced the results.

Authors’ response: We acknowledged in the Methods section under “Procedures for tru-cut biopsy and ascitic fluid cytology” that sampling 20mL is our institutional protocol- Lines 99-100, Page 5. We noted studies that recommended collecting larger volumes up to 200mL and taking multiple samples on different days, as well as highlighted how our smaller volume of 20ml may have influenced our results: “They reported that an ascitic fluid volume of 60-100 mL, and three sequential collections on different days of a patient’s admission provided the best diagnostic accuracy of up to 99.5%.18 Zhang et al recommended 200 mL as the optimum minimum volume for confirming malignant ascites.15 In our study, 20 mL of ascitic fluid was collected only once for cytological diagnosis. This may possibly explain the lower diagnostic accuracy recorded in our study.”- Lines 240-246, Pages 12. Thank you.

Comment 12: The results are presented neatly, with tables to compare diagnostic values of both techniques.

Authors’ response: Thank you.

Comment 13: There is good reporting of the complication rate, which adds to the clinical applicability of the findings.

Authors’ response: Thank you.

Comment 14: However, subgroup analysis (e.g., by tumour type or stage) could be informative, as high-grade ovarian cancers predominated the cohort.

Authors’ response: Histological subtypes of ovarian cancers, which predominated this study, have been provided in Table 2- Lines 177-179, Page 9. Thank you.

Comment 15: The discussion is excellent in contrasting the findings with other literature and highlighting the benefits of tru-cut biopsy versus ascitic cytology.

Authors’ response: Thank you.

Comment 16: The authors have referred to the limitation of single-center study, small sample size, and retrospective design, which is welcome.

Authors’ response: Thank you.

Comment 17: Additional discussion regarding the clinician's practical rule of thumb of how to utilize tru-cut and how frequently just cytology would suffice would be helpful.

Authors’ response: We discussed the clinician’s practical rule of thumb of how to utilize tru-cut and cytology as follows: “Tru-cut biopsy demonstrated higher overall diagnostic accuracy, sensitivity, and negative predictive value, making it the preferred modality when a definitive tissue diagnosis is required. However, ascitic fluid cytology retains a critical role in the diagnostic pathway. Its high specificity and positive predictive value, combined with ease of performance, absence of complications, and good adequacy rates, make it especially useful when biopsy is not feasible or in settings with limited resources. Importantly, a positive cytology result reliably confirms malignancy and can expedite treatment decisions, particularly in patients with advanced ovarian cancer where surgery is not immediately indicated.”- Lines 296-304, Pages 14-15. Thank you.

Comment 18: The conclusion answers the research question appropriately.

Authors’ response: Thank you.

Comment 19: The authors summarize the key findings clearly, but they may consider adding a forward-looking statement on how these findings could guide diagnostic protocols in gynecologic oncology.

Authors’ response: We highlighted in the Conclusion that: “Thus, while tru-cut biopsy may be superior in diagnostic performance, ascitic fluid cytology remains a relevant and valuable first-line diagnostic option in gynaecologic oncology, especially in settings where biopsy is unavailable or unfeasible.”- Lines 304-307, Page 15.

Comment 20: Overall Evaluation: This is a well-conducted and clinically relevant study, which compares two important diagnostic examinations in gynecologic oncology. It is well written, well-reasoned, and adds valuable evidence to the literature. With minor revision to further bolster the clinical utility and defend some of the methodological choices, the manuscript is publishable.

Authors’ response: Thank you for your very valuable comments. The manuscript has been revised in line with your very valuable comments and suggestions.

Reviewer 2:

Comment 1: For the tru-cut biopsy procedure, was attention paid to patients’ pre-procedural platelet counts and the use of anticoagulants, particularly heparin? The Statistical Analysis section should be expanded in greater detail. Was a power analysis performed?

Authors’ response: A complete blood count, including platelet count, is routine in the evaluation of patients with gynaecologic tumours in our institution. However, information on attention to patients’ pre-procedural platelet counts and the use of anticoagulants could not be obtained from the patients’ case records, being a retrospective study.

This statement has been added to the Statistical Analysis sec

---

## [Editor Report · Decision Letter 1]

3 May 2026

Transvaginal tru-cut biopsy versus ascitic fluid cytology in gynaecologic tumours: a comparative diagnostic study

PONE-D-25-37903R1

Dear Dr. Ubom,

We’re pleased to inform you that your manuscript has been judged scientifically suitable for publication and will be formally accepted for publication once it meets all outstanding technical requirements.

Kind regards,

Manasa Varra

Academic Editor

PLOS One

Additional Editor Comments (optional):

Dear Akaninyene Eseme Bernard Ubom,

I am happy to inform you that all the reviewer's comments have been addressed, and the necessary incorporations have been made in the Revised Manuscript Number PONE-D-25-37903R, entitled,

Transvaginal tru-cut biopsy versus ascitic fluid cytology in gynaecologic tumours: a comparative diagnostic study.

Therefore, I recommend that the same may be accepted for publication in PLOS One.

Regards,

Dr.Manasa V
---

## [Editor Report · Acceptance letter]

PONE-D-25-37903R1

PLOS One

Dear Dr. Ubom,

I'm pleased to inform you that your manuscript has been deemed suitable for publication in PLOS One. Congratulations! Your manuscript is now being handed over to our production team.

Kind regards,

on behalf of

Dr. Manasa Varra

Academic Editor

PLOS One